# Interactive Design of Gallery Walls via Mixed Reality

## ABSTRACT

We present a novel interactive design tool that allows users to create and visualize gallery walls via a mixed reality device. To use our tool, a user selects a wall to decorate and chooses a focal art item. Our tool then helps the user complete their design by optionally recommending additional art items or automatically completing both the selection and placement of additional art items. Our tool holistically considers common design criteria such as alignment, color, and style compatibility in the synthesis of a gallery wall. Through a mixed reality device, such as a Magic Leap One headset, the user can instantly visualize the gallery wall design in situ and can interactively modify the design in collaboration with our tool's suggestion engine. We describe the suggestion engine and its adaptability to users with different design goals. We also evaluate our mixed-reality-based tool for creating gallery wall designs and compare it with a 2D interface, providing insights for devising mixed reality interior design applications.

## ACM Classification Keywords

H.5.m. Information interfaces and presentation (e.g., HCI)

## Author Keywords

Design interfaces; mixed reality; spatial computing

## INTRODUCTION

The advent of mixed reality devices (e.g., Microsoft Hololens, Magic Leap One) gives rise to new and exciting opportunities for spatial computing. The superior immersive visualization and interaction experience provided by these devices promises to change the way interior design is performed as they allow users to instantly preview and modify designs in real spaces.

Interior design has historically been a costly and time-intensive process. The conventional design process involves contemplating f abric swatches and inspirational photos, as well as talking to a designer. A professional designer may make use of 3D modeling software to preview a design on screen through sophisticated manual operations. The designer's client must then mentally translate what they see on the 2D screen to how the design may look in a real living space. Without convenient means for visualizing and modifying designs, the design process can be tedious and nonintuitive. Such limitations restrict the ability of general users to

*Graphics Interface 2020* 21–22 May 2020, Toronto, Ontario, Canada

DOI: http://dx.doi.org/10.475/123_4

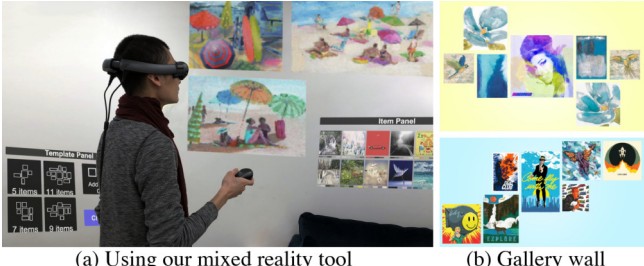

(a) Using our mixed reality tool for designing a gallery wall

(b) Gallery wall designs

Figure 1: (a) A user wearing a Magic Leap One headset designs a gallery wall using our tool. The figure shows what the user sees in mixed reality while designing: the control panels and his gallery wall design overlaid on the real wall. (b) Some gallery walls designed by users with our tool.

engage in interior design even though they may have creative ideas.

In our work, we attempt to address these challenges by devising an interior design tool leveraging the visualization and interaction capabilities of the latest consumer-grade mixed reality devices. Since interior design is a broad area, as an early attempt to investigate such design applications based on mixed reality, we particularly focus on the design of gallery walls, which are common for decorating interior spaces such as living rooms, hotel lobbies, galleries, and other interior spaces. Figure 1 shows a user designing a gallery wall using our tool.

A **gallery wall** refers to a cluster of wall **art items** artistically arranged on a wall. A gallery wall commonly contains a **focal item** near the center of the arrangement that sets the tone for the overall design. The other items are called **auxiliary items** and are placed around and with reference to the focal item. The auxiliary items are generally compatible with the focal item in terms of color and style. These definitions follow the conventions used by designers in creating a gallery wall design using a traditional workflow.

Our tool is suitable for novice users. Users are able to directly visualize how the gallery wall design will look on the real wall while interactively creating the design. The visualization and user-interaction components of our tool keeps the user in the loop of the design process, allowing quick exploration of desirable designs through trial-and-error.

Furthermore, by comparing the color and style compatibility between different art items, our suggestion engine can let the user quickly browse through many desirable design suggestions automatically generated by our tool, hence saving the manual and mental efforts involved in browsing through a large database of wall art items.

We make the following major contributions in this work:

- Based on interviews with professional designers, we devise a computational approach for facilitating and automating the design of gallery walls, which enables a novel mixed reality interactive design tool.

- We demonstrate how mixed reality technology, which bridges the gap between real-world scene knowledge and design suggestions computed in a virtual setting, can be adopted for interior design. In our case, we particularly demonstrate how such an approach can be applied for designing gallery walls.

- We conduct experiments to evaluate the user experience and performance of using our novel tool for gallery wall design. We also conduct a perceptual study to evaluate the quality of the gallery wall designs created by users with our tool.

We believe these contributions will inspire future research in creating mixed reality interfaces for interior design.

### RELATED WORK
To the best of our knowledge, there is no existing work on using mixed or augmented reality for gallery wall design. Regardless, we briefly review the existing research work and commercial tools relevant to our problem domain.

### Extended Reality for Interior Design
Companies have been exploring the use of virtual, augmented, or mixed reality technologies for creating and visualizing interior design. Matterport uses a 3D camera to capture the color and depth of real-world living spaces, which can be visualized in 3D by users wearing a virtual reality headset. Such an approach finds promising applications for virtual real estate tours. On the other hand, roOomy provides virtual staging services, enabling users to see previews of interior designs via augmented reality devices showing virtual furniture objects overlaid on a real scene. Furniture retailers such as Wayfair also develop virtual and augmented reality experiences with capabilities such as customizing the design of outdoor spaces with furnishings and décor.

Several companies provide web interfaces or mobile applications for designing gallery walls. For example, Shutterfly allows users to upload their photos and arrange them using preset layouts provided by the company. Art.com provides a mobile application that allows users to select individual art items or preconfigured gallery wall layouts from their large wall art collection and visualize them via augmented reality on a mobile device.

Recently, several researchers proposed using extended reality for interior design [2, 10, 14, 22, 31, 30]. Zhang et al. [31] proposed an approach to add furniture items and relight scenes on a RGBD-enabled tablet. Yue et al. [30] developed a mixed reality system which allows users to efficiently edit a scene for applications such as room redecoration. Virtual content needs to be adapted to fit the current scene. Nuernberger et al. [14] devised a technique to align virtual content with the real world. Chae et al. [2] proposed a space manipulation technique for placing distant objects by dynamically squeezing surrounding spaces in augmented reality. Lindlbauer and Wilson [10] showed how to manipulate space and time in augmented reality.

Compared to the existing approaches, our tool not only uses extended reality technologies for visualizing gallery wall designs, but also reasons about the spatial and color compatibility. Our tool simplifies the design process by suggesting desirable combinations and placements of art items, taking the color of the wall into account.

### Automated Layout Design
Recently, layout design automation has received much research attention. There are previous efforts on automatic 2D graphics layout design [16, 15], poster design [18], website design [17], magazine covers [8], and photo collages [19, 24]. We focus on reviewing automatic scene layout design works.

Merrell et al. [12] proposed an interactive tool for furniture layout design based on interior design guidelines, while Yu et al. [28] devised an optimization framework for automatic furniture layout design. Fisher et al. [6] characterized structural relationships in scenes based on graph kernels and later proposed an example-based approach [5] for synthesizing 3D object arrangements for modeling partial scenes. More recently, Wang et al. [21] applied deep convolutional priors for indoor scene synthesis, while Weiss et al. [23] proposed a physics-based approach for fast and scalable furniture layout synthesis. Such works are mostly focused on the geometrical aspects of populating spaces with furniture. Complementary to this line of work, Chen et al. [3] created a tool called *Magic Decorator* for automatically assigning materials for objects in indoor scenes. Xu [26, 27] proposed a tool for layout beautification and arrangement. However, none of these approaches considers the stylistic arrangement of wall art items in a scene. We propose a novel approach for modeling common factors such as colors, spatial relationships, and semantic compatibility among art items to generate desirable gallery walls, which could complement the existing automated interior design approaches.

We also note that recently CAD software companies such as Autodesk are applying generative design for automating layout synthesis [13]. Along with this line of work, we believe our generative design tool for semi-automating gallery wall design will also find good practical uses.

### Interactive Scene Modeling
Typically, users want the ability to control, modify, and visualize the design during the design process so that they can infuse their personal stylistic preferences in their designs. As such, interactive modeling tools play an important role in the design process. There is a large body of work on interactive modeling tools. We review recently proposed scene modeling interfaces.

Along the direction of suggestive interfaces for scene modeling, Yu et al. proposed a suggestive interface called *ClutterPalette* [29] that uses object distribution statistics learned from real-world scenes for suggesting appropriate furniture

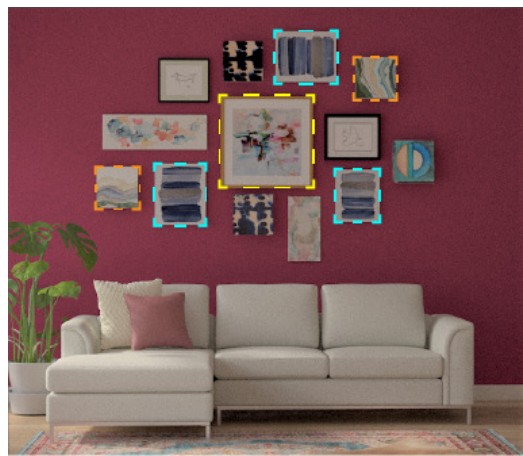

Figure 2: A gallery wall created by a designer using a conventional workflow. The focal item (in yellow), as well as a diagonal pair (in orange) and a triangular group (in cyan) of auxiliary items are highlighted.

items to add to a scene. Matthew et al. [4] devised a context-based search engine for 3D furniture models to add to a scene.

Another line of work focuses on providing users with easy controls for creating objects while modeling a scene. As humans are accustomed to drawing sketches, a promising approach is to devise sketch-based interfaces for modeling scenes. For example, Xu et al. [25] proposed a sketch-based interface for retrieving and placing 3D models in scenes. Recently, Li et al. [9] devised an interface called *SweepCanvas* that allows users to perform 3D prototyping on an RGB-D image of a partial scene by sketching. Users can conveniently create virtual objects overlaid on top of the point cloud of a real scene.

Compared to the existing interfaces, we propose a novel mixed reality-based interface for designing gallery wall layouts *in situ*, with the user seeing the design overlaid on top of the target wall. Seeing how the generated design fits into the real world, the user can easily modify the design by a few intuitive operations. We demonstrate in our evaluation experiments that our tool can allow not only designers but also novice users to quickly generate desirable gallery wall designs.

### INTERVIEW WITH DESIGNERS ON WORKFLOW

To devise a computational approach and a practical tool for designing gallery walls, we interviewed 4 professional designers from a large furnishings and décor company to better understand the way professional designers create gallery walls under current practice. Each of the designers has at least 5 years of interior design or staging experience and has designed dozens of gallery walls in their professional capacity.

Figure 2 shows a gallery wall created by a designer. The general process of designing a gallery wall is as follows:

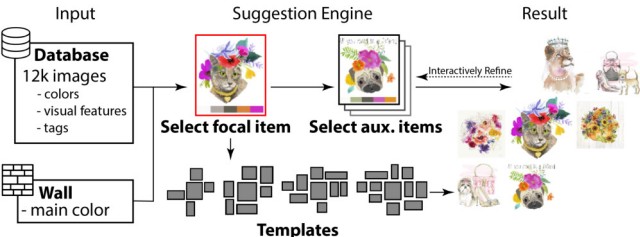

Figure 3: System overview. Taking a database of art items and a wall as input, our tool suggests a focal art item as the starting point for designing a gallery wall. The user may apply a template to generate an initial design, and then interactively refines the design with the help of our suggestion engine.

1. The designer first observes the style of the room, particularly paying attention to the colors of the wall and the furniture objects that the designer would like to decorate around.

2. The designer explores a database containing many art items and chooses a focal art item, which is to be placed near the center of the gallery wall and serves as the reference for placing other auxiliary art items. The color of the focal art item should be compatible with the wall color.

3. The designer selects and adds the auxiliary art items to the gallery wall. The colors and styles of these art items should contain some variety, yet they should all be compatible with those of the focal art item.

4. The designer lays out the auxiliary art items around the focal art item nicely. One common consideration is balance: pairs of similar art items are placed at opposite sides of the focal art item.

5. The designer refines the locations of the art items to achieve proper spacing and alignment between items.

We devise our computational approach based on the above observations to automate the conventional design process. Typically, in creating a gallery wall for a common scene like a living room, it takes about 20 minutes for the designer to select compatible wall art items and another 20 minutes to arrange the items into a good layout.

### SYSTEM OVERVIEW

Our tool is realized as an application that runs on the Magic Leap One mixed reality headset. Figure 1 shows a user designing a gallery wall using our tool. The user wears the headset when using our application to create a gallery wall.

The display on the mixed reality headset shows a user interface as well as the virtual gallery wall design overlaid on the real wall, such that the user can instantly preview how the gallery wall design will look on the real wall while designing. The user interacts with the user interface via a handheld controller (e.g., a Magic Leap One's Control) to perform operations such as selecting and dragging.

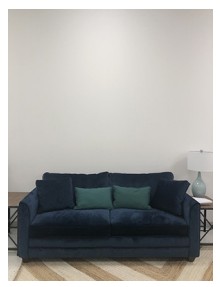 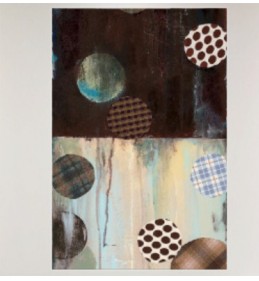 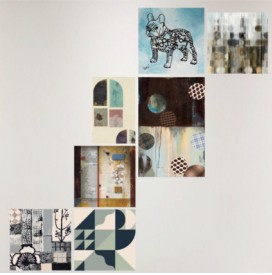 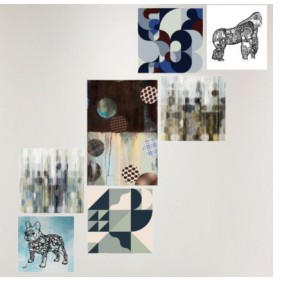 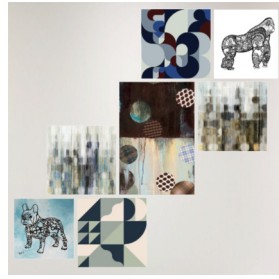

| (a) Input Wall | (b) Focal Item Selection | (c) Initialization | (d) Interactive Refinement | (e) Result |

Figure 4: Designing a gallery wall with our tool. (a) The input empty wall. (b) The user selects a focal item. (c) The user applies a template for initializing the design. (d) The user modifies the design interactively. (e) The finished gallery wall design.

Figure 3 shows an overview of our tool. It consists of two major components: a suggestion engine for generating design suggestions and a user interface for modifying the gallery wall design. The tool is connected to a large database of wall art items (pictures) from which suitable art items are automatically retrieved and suggested to the user.

### Workflow

Figure 4 illustrates the typical user workflow while using our tool for designing a gallery wall. The input is a wall color. The output, which can be reached with as few as three user decisions, is a gallery wall design that goes well with the wall color. Our tool achieves color compatibility by using the wall color to retrieve candidate focal art items whose colors are compatible with the wall color. A user selects a focal art item from these suggestions, picks a focal item size from among offered art sizes, and either chooses a gallery wall template to launch the synthesis of a gallery wall design or browses recommended auxiliary items.

The selection of auxiliary items, whether human-selected or template-selected, begins with the generation of a color palette based on the colors of the wall and the selected focal item. Based on this color palette and the style of the focal item, the system then retrieves from the database a selection of auxiliary art items that are either presented to the user as options or automatically incorporated into a layout.

After the initialization, the user can interactively modify the gallery wall design via operations such as dragging-and-dropping and selecting in the 3D space using a handheld controller. For example, the user can move, resize, replace, add, or remove any art items. Our tool also provides semi-automatic operations to help the user refine the design, for example, by performing automatic snapping of the art items. The design session ends when the user is satisfied with the gallery wall.

### Art Items Data

**Database.** Our tool is built upon a suggestion engine which helps users browse and make selections from a large database of wall art items while designing their gallery walls. The database, which contains 12,000 wall art items created by

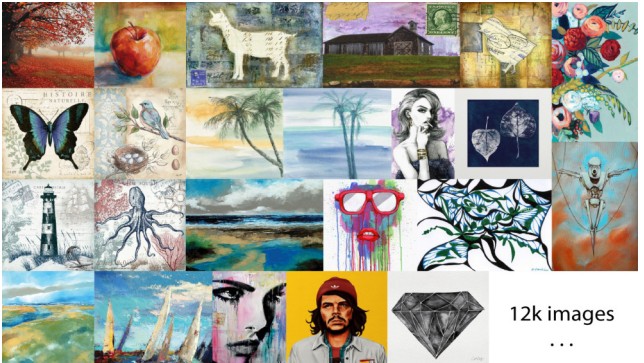

Figure 5: Samples of wall art items in our database, which contains more than 12,000 art items.

professional artists, belongs to a company specialized in furniture and interior design business. Figure 5 depicts some example art items. The art items are mostly paintings and stylized photographs. Each art item comes with 1 to 5 realistic dimensions (from small to large) whose real replicate can be ordered and put on a real wall.

**Annotations.** To compute the compatibility between different art items for making suggestions, each item is annotated with:

1. **Colors.** The 5 most dominant colors of the art item are extracted by the k-means clustering algorithm and stored.

2. **Visual Features.** 256 visual feature values are computed by a convolutional neural network, which encode the visual style of the art item. We trained a Siamese Network to perform such feature extraction using the aforementioned database containing many art items. The network takes image pairs as input, where a positive pair consists of images of items from the same category and a negative pair consists of images of items from different categories. Each input image is processed by a modified Inception-ResNet [20], whose last layer is changed to a fully connected layer, resulting in a 256-dimensional vector as the output. The network was trained to minimize the contrastive loss [7] of the vectors (embeddings) of the input image pairs.

| Tags: Style | | | | |
|---|---|---|---|---|
| American Traditional | Asian Inspired | Beachy | Bohemian & Bold | Eclectic Modern |
| Cabin / Lodge | Coastal | Cottage / Country | Cottage Americana | Eclectic |
| French Country | Glam | Global Inspired | Industrial | Mid-Century Modern |
| Modern & Contemporary | Modern Farmhouse | Modern Rustic | Nautical | Ornate Glam |
| Ornate Traditional | Posh & Luxe | Rustic | Scandinavian | Sleek & Chic Modern |
| Traditional | Tropical | | | |

| Tags: Subject | | | | |
|---|---|---|---|---|
| Abstract | Abstract | Bath & Laundry | Buildings & Cityscapes | Cities & Countries |
| Entertainment | Fantasy & Sci-Fi | Fashion | Floral & Botanical | Food & Beverage |
| Geometric | Humor | Inspirational Quotes & Sayings | Landscape & Nature | Maps |
| Nautical & Beach | People | Spiritual & Religious | Sports & Sports Teams | Transportation |

| Tags: Color | | | | |
|---|---|---|---|---|
| Beige | Black | Blue | Brown | Chrome |
| Clear | Gold | Gray | Green | Orange |
| Pink | Purple | Red | Silver | Tan |
| White | Yellow | | | |

Table 1: The tags in our database, which are manually assigned to the art items by professional designers. The database consists of 3 categories, namely, style (27 tags), subject (20 tags), and color (17 tags).

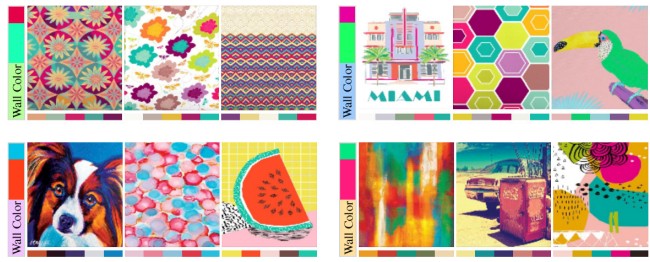

Figure 6: Examples of focal items retrieved by the suggestion engine based on different wall color schemes.

3. **Tags.** Each item also carries tags manually specified by designers of the company. (See Table 1)

Similarity between two art items is computed based on the L2 distance between their annotation vectors: the smaller the distance, the more similar the two items are. We find these annotations useful in devising our suggestion engine as they characterize the art items and are also common criteria used by designers for comparing art items. By formulating our scoring functions using these three types of annotations, we are able to devise an interface that allows the user to flexibly apply filters using a subset or all three types of annotation to retrieve relevant art item suggestions, making it easy for the user to browse through the large database of art items.

**TECHNICAL APPROACH**

We provide details for our design suggestion engine. There are two major components: art items suggestion and templates. By using these components, the user can quickly browse through the database of art items and select items that fit with the wall, as well as obtaining a decent spatial arrangement of the items as an initialization of their design.

**Wall Plane and Color**

Akin to the conventional workflow for designing a gallery wall, our approach starts with considering the wall color. The user wears the Magic Leap One headset and faces the target wall to be decorated. The wall plane is detected and extracted based on the headset's built-in functionality. The user can manually specify the wall color via a color picker in the user interface, or by using the headset's camera to take a picture of the wall whose average color is taken as the wall color.

Based on the wall color, a neighbor color within $\pm(60° \text{ to } 90°)$ of the wall color is randomly selected from the HSV circular color space. The complementary color of the neighbor color (180° from the neighbor color in the HSV circular color space) is also selected. Figure 7 shows an example. The wall color is used as the basis for retrieving other colors for suggesting relevant art items.

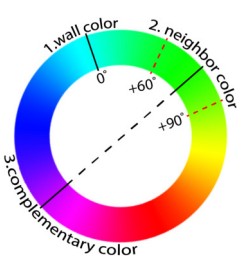

Figure 7: Wall's color palette example.

**Suggested Focal Items**

Our goal in this step is to retrieve and suggest a list of art items from the database as candidate focal art items for the user. Figure 6 shows some suggested focal art items based on different wall colors.

To achieve this, the wall color, neighbor color and complementary color then form a 3-color palette based on which compatible focal art items are selected. The wall compatibility score $S_{\text{foc}}$ of a candidate focal art item $\phi$ is defined as:

$$S_{\text{foc}}(\phi) = 1 - \frac{1}{9} \sum_{\mathbf{c}_w \in C_w} \min\{d(\mathbf{c}_w, \mathbf{c}_\phi) \mid \mathbf{c}_\phi \in C_\phi\}, \quad (1)$$

where $C_w$ is a set containing the wall color, neighbor color and the complementary color in the HSV space; $C_\phi$ is a set containing the 5 dominant colors of the candidate focal art item $\phi$ in the HSV space. This scoring function evaluates how close the candidate focal art item $\phi$'s color palette is with respect to the wall's color palette. The closer they are, the higher the wall compatibility score.

Note that $\mathbf{c}_w, \mathbf{c}_\phi \in \mathbb{R}^3$ are colors in the HSV space. $d(.)$ is a distance metric function to project the two colors $\mathbf{c}_w$ and $\mathbf{c}_\phi$ into the HSV cone and to compute the squared distance between them in that cone [1]:

$$
\begin{aligned}
d(\mathbf{c}_w, \mathbf{c}_\phi) = {} & (\sin(H_w)S_w V_w - \sin(H_\phi)S_\phi V_\phi)^2 \\
& + (\cos(H_w)S_w V_w - \cos(H_\phi)S_\phi V_\phi)^2 \\
& + (V_w - V_\phi)^2,
\end{aligned}
$$

where $H \in [0, 2\pi)$, $S \in [0, 1]$, and $V \in [0, 1]$ are the HSV channel values. The range of $d(\mathbf{c}_w, \mathbf{c}_\phi)$ is $[0, 3]$. Equation (1) sums up the differences of 3 pairs of colors, hence a normalization of 9 is used.

Our approach computes the compatibility scores for all the art items in the database. The top-20 art items are retrieved and displayed in order of the compatibility scores with the highest-scoring item shown first. The user is supposed to select a focal art item from the list of suggested art items. However, if needed, the user can also explore the database to select any other art item as the focal art item using the Item Panel which we describe in a later section.

**Suggested Auxiliary Art Items**
Akin to the conventional gallery wall design approach, the selected focal art item serves as a reference for the suggestion engine to suggest other compatible, auxiliary art items to add to the gallery wall design. To retrieve auxiliary art items from the database as suggestions, an overall compatibility score $S_{aux}$ is computed for each candidate auxiliary art item, which evaluates the style and color compatibility between the auxiliary art item and the selected focal art item:

$$
S_{aux}(\phi) = w_c S_{aux}^c(\phi) + w_s S_{aux}^s(\phi), \tag{2}
$$

where $w_c$ is the weight of the color compatibility score $S_{aux}^c$ and $w_s$ is the weight of the style compatibility score $S_{aux}^s$.

*Color Compatibility Score:* A candidate auxiliary art item $\phi$ has a high color compatibility score $S_{aux}^c$ if its colors are close to the dominant colors of the selected focal art item. Specifically, the color compatibility score of auxiliary art item $\phi$ is defined as follows:

$$
S_{aux}^c(\phi) = 1 - \frac{1}{15} \sum_{\mathbf{c}_f \in C_f} \min\{d(\mathbf{c}_f, \mathbf{c}_\phi) \mid \mathbf{c}_\phi \in C_\phi\}, \tag{3}
$$

where $C_f$ and $C_\phi$ are respectively sets containing the 5 dominant colors of the selected focal art item and of the auxiliary art item $\phi$. $\mathbf{c}_f, \mathbf{c}_\phi \in \mathbb{R}^3$ are colors in the HSV space. As Equation (3) sums up the differences of 5 pairs of colors, and each difference has a range of $[0, 3]$, a normalization of 15 is used.

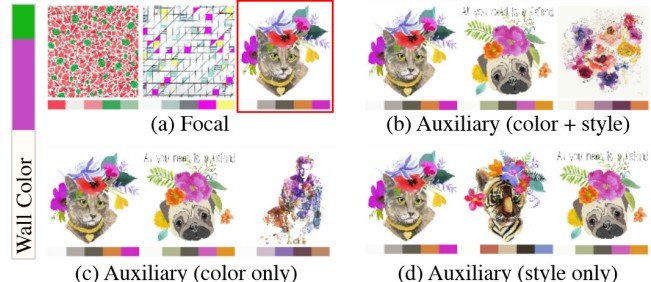

(a) Focal        (b) Auxiliary (color + style)

(c) Auxiliary (color only)      (d) Auxiliary (style only)

Figure 8: Art item suggestions. Based on the wall's color palette shown on the left, (a) several color compatible focal art items are suggested. Based on a selected focal art item (highlighted in red), (b) suggested auxiliary art items compatible in color and style. Auxiliary art items suggested by considering (c) only color or (d) only style.

This scoring function evaluates how close the auxiliary art item $\phi$'s color palette is with respect to the selected focal art item's color palette. The closer they are, the higher the score is.

*Style Compatibility Score:* A candidate auxiliary art item $\phi$ has a high style compatibility score $S_{aux}^s$ if its visual feature vector is close to the selected focal art item's visual feature vector. The style compatibility score of auxiliary art item $\phi$ is defined as follows:

$$
S_{aux}^s(\phi) = 1 - \frac{1}{\sqrt{n}} \|\mathbf{v}_f - \mathbf{v}_\phi\|, \tag{4}
$$

where $\mathbf{v}_f, \mathbf{v}_\phi \in \mathbb{R}^n$ are the $n$-dimensional visual feature vectors of the focal item and auxiliary art item $\phi$ computed by the convolutional neural networks. The size of the dimension in our database is 256.

Overall, the compatibility score is computed for each art item in the database. The user interface displays the top-20 art items sorted in descending order of their compatibility scores as auxiliary art item suggestions. To provide flexibility in retrieving suggestions, our user interface allows the user to turn on and off the consideration of the color or the style compatibility score, which correspond to setting $w_c$ or $w_s$ as 1 or 0. Figure 8 shows an illustration. The user can also select which of the 5 dominant colors $C_f$ of the selected focal art item to consider in computing the color compatibility score.

**Templates**
To allow the user to quickly generate an initial gallery wall design based on a focal item, our tool provides preset templates that the user can choose from and apply. Figure 12 shows some example templates that our tool provides. These templates encode spatial relationships of art items that are commonly applied by gallery wall designers. A template arranges groups of auxiliary art items symmetric about and around the focal art item, akin to the gallery wall designs created by the conventional design workflow. Figure 9(b) shows the 4 templates (with 5, 7, 9, and 11 items) that we provide with our tool in our experiments. These templates resemble

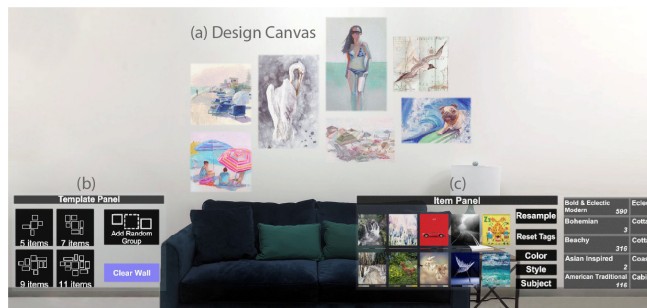

(a) Design Canvas

(b) Template Panel

(c) Item Panel

Figure 9: User interface of our tool via which a user wearing a mixed reality headset visualizes and designs a gallery wall. It consists of three components: (a) Design Canvas; (b) Template Panel; and (c) Item Panel.

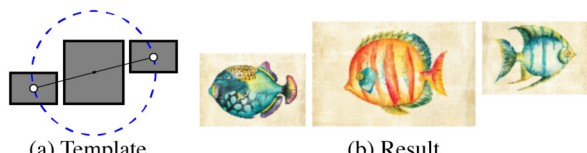

(a) Template                    (b) Result

Figure 10: Applying a template to generate a gallery wall design. (a) A template and its tree structure. The auxiliary items are placed symmetrically about the focal item at the center. (b) A gallery wall created by applying this template.

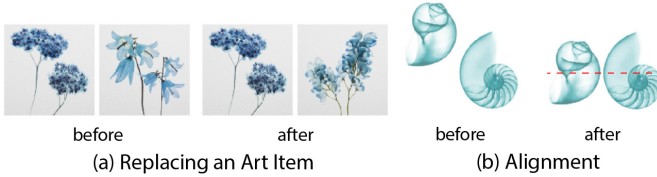

before          after          before          after

(a) Replacing an Art Item          (b) Alignment

Figure 11: The user can interactively modify a design in mixed reality using the functionalities of our user interface.

the common patterns used by designers as we learned from the interview. Auxiliary items within a group have compatible color and style by default (Equation (2)).

**Initializing a Gallery Wall Design.** Figure 10 illustrates how to apply a template to generate a gallery wall design. According to the layout of the chosen template, starting from the root (the focal art item), our approach inserts auxiliary art items which are compatible with the focal art item, group by group. Specifically, the items are added on the circumference of a circle centered at the focal item with a random radius $r \in [0.5d, 3d]$, where $d$ is the diagonal length of the focal item. A pair of two auxiliary items would be added at the opposite sides of the circle. A group of three items would be added at the vertexes of a randomly-oriented equilateral triangle circumscribed by the circle. The gallery wall design generation finishes as all groups of auxiliary art items have been placed. The generated design is taken as an initial design based on which the user can modify interactively.

**Spatial Refinement.** Our approach refines the spatial relationships between the items after the initialization and after every user interaction with the gallery wall design such as adding an item, removing an item, and moving an item.

*Snapping:* To keep the gallery wall design compact, by default, all auxiliary items steer toward the focal item at the center while they maintain a certain minimum space between each other to avoid overlapping.

*Alignment:* To keep the gallery wall design neat and uncluttered, by default, our approach aligns neighboring art items either horizontally or vertically by their edges so long as the alignment does not cause overlapping. (Figure 11(b))

We include more examples of interactively modifying actions in the supplementary video.

### USER INTERACTION

Figure 9 shows the user interface of our tool which is displayed in mixed reality. It consists of three components: a) the *Design Canvas* where the user can interactively modify the current gallery wall design visualized on the real wall; b) the *Template Panel* where the user can select and apply a pre-

set template for synthesizing an initial gallery wall design; and c) the *Item Panel* where the user can retrieve art items from the database by specifying different criteria. Each of the components allows users to refine a gallery wall design conveniently and desirably. We describe them in the following.

### Design Canvas

The design canvas visualizes and overlays the current gallery wall design on the real wall via the mixed reality headset's display. It also provides support for interactively adjusting both art items and the gallery wall layout:

- **Add.** The user selects an art item in the current design and retrieves several art items from the database that are compatible in terms of dominant colors, visual features and tags, which he can add to the current design.

- **Replace.** The user replaces an art item with another compatible art item from the database. (Figure 11(a))

- **Move.** The user moves an art item by dragging it.

- **Resize.** The user chooses another size for an art item.

- **Remove.** The user removes an art item.

### Template Panel

The Template Panel allows the user to quickly generate an initial gallery wall design with items decently placed. It provides a number of preset templates that the user can apply to synthesize a gallery wall design based on a selected focal art item. It also provides other functionalities to enable automatic refinement of the spatial layout of the current design. A list of functionalities supported:

- **Apply a Template.** Based on a placed focal art item, the user applies a template to synthesize a gallery wall design.

- **Add Random Group.** Our tool automatically adds a group of 2 or 3 auxiliary art items, which are compatible with the

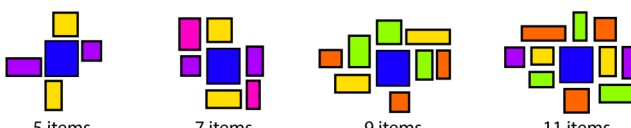

Figure 12: The templates we use in our system. The blue item represents the focal item. The colored items represent the groups of 2 or 3 auxiliary items.

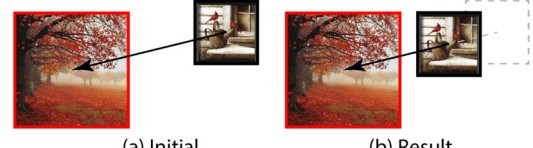

(a) Initial          (b) Result

Figure 13: Snapping example. (a) The user drags an art item, which is (b) snapped toward the center of its neighbor item until a minimum spacing between the two is reached.

focal art item, to the current design. The group of auxiliary items are symmetric about the focal item (Figure 12).

- **Align All.** The user triggers our tool to align all the art items with respect to each other. The alignment is done along the horizontal (left, right, or center) or vertical (top, bottom, or center) direction. This functionality comes in handy because it could be tiring and difficult for users to preform multiple precise adjustments in the 3D space [11] using a handheld controller.

- **Snap.** If enabled, an art item is snapped to its neighbor item as the user drags the art item around, i.e., it will steer toward the center of its neighbor item until a minimum spacing between the two items is reached (Figure 13).

- **Clear Wall.** All the art items are removed from the design.

**Item Panel**

The Item Panel is connected to the suggestion engine and the database of art items. Its primary function is to display a relevant list of art items that the user can add to the gallery wall design as a focal art item or auxiliary art items. The panel contains buttons that the user can click to set the criteria for retrieving relevant art items from the database. For example, the user can select whether to use color, or style, or both as criteria for determining compatibility between items. The user can also select which color(s) out of the 5 dominant colors of the focal art item to use for determining color compatibility. A list of functionalities supported:

- **Update Wall's Color Palette.** Our tool re-generates the neighbor and complementary colors of the wall's color palette.

- **Find Focal.** Based on the wall's color palette, our tool retrieves several of compatible art items as focal art item suggestions (according to equation (Equation 1)).

- **Find Auxiliary.** The user selects criteria (e.g., colors, visual features, tags) based on which our tool retrieves 20 compatible art items as auxiliary art item suggestions. By default, the art items are sorted by their compatibility scores.

### USER EVALUATION

We developed our tool using C# on the Unity Game Engine installed with the Magic Leap Lumin SDK. We deployed our tool onto a Magic Leap One headset which we used for our user evaluation experiments.

**User Groups.** We recruited two different groups of users to evaluate our tool.

*Group 1:* The first group was recruited to evaluate the user experience of designing a gallery wall using our mixed reality interface based on Magic Leap One versus using a 2D interface which mimics a traditional design tool on a laptop. We recruited 17 participants, who are the employees of a company, consisting of 12 males and 5 females, aged from 20 to 45, the average age was 32. All participants did not have experience using the Magic Leap One headset. Each participant designed 2 gallery walls, under **Condition MR** and **Condition 2D** in a random order.

*Group 2:* The second group was recruited to evaluate the user experience of designing a gallery wall without and without the template functionality. We recruited 24 participants, who are the college students, consisting of 16 males and 8 females, aged from 19 to 24 the average age was 22. Each participant designed 2 gallery walls under **Condition 2D** and **Condition 2DNT** in a random order.

**Conditions.** We asked the participants to create gallery wall designs. The goal of each task was to design a gallery wall that fits with a living room with a pale gray wall and a blue sofa as shown in Figure 9.

- *Condition MR:* The participant used our tool delivered through a mixed reality interface to create a gallery wall.

- *Condition 2D:* The participant used our tool delivered through a 2D interface to create a gallery wall.

- *Condition 2DNT:* The participant used our tool delivered through a 2D interface to create a gallery wall with *no template functionality*. That is, the "Apply a Template" and "Add Random Group" buttons under the Template Panel were disabled.

Note that, both **Condition MR** and **Condition 2DNT** used a background image showing a pale gray wall and a blue sofa. We include the 2D interface we used for the user evaluation in the supplementary material.

**Procedure.** Before each task, we briefed and trained the participant in creating a gallery wall design using our tool. We asked the participant to follow a 5-minute tutorial which guided him how to use our tool to create a gallery wall design step by step. Note that we showed the participant the tutorial whether he was tasked with creating a gallery wall using the mixed reality interface (Condition MR) or the 2D interface (Condition 2D). The participant could ask any question about the user interface to make sure he was familiar with using it.

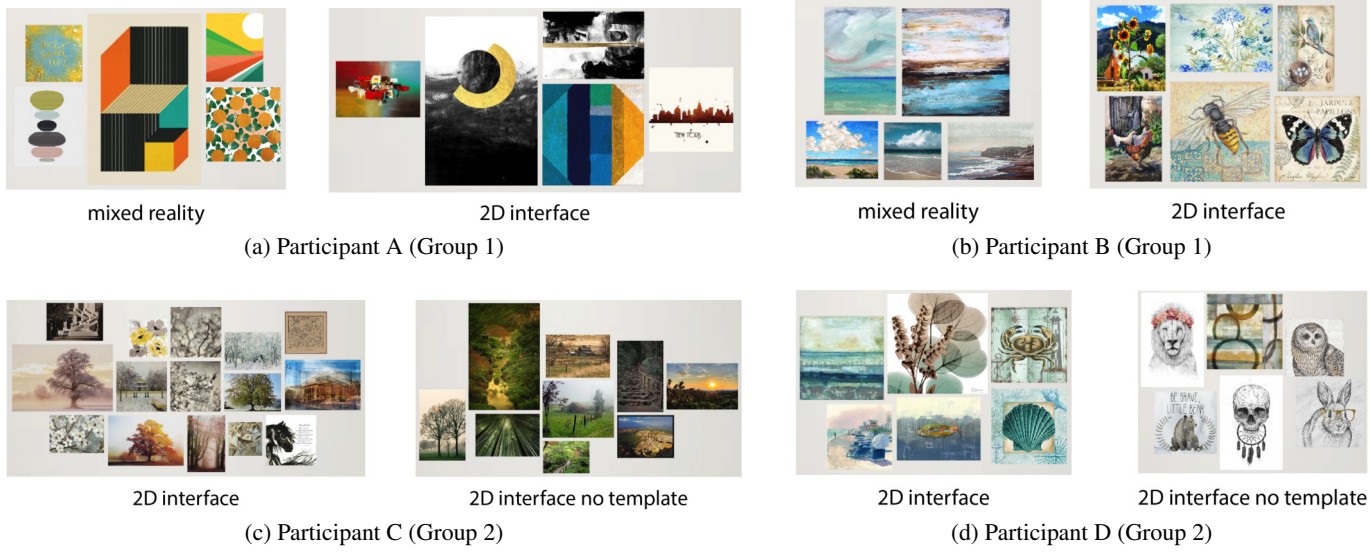

| mixed reality | 2D interface |
|---|---|
| (a) Participant A (Group 1) | |

| mixed reality | 2D interface |
|---|---|
| (b) Participant B (Group 1) | |

| 2D interface | 2D interface no template |
|---|---|
| (c) Participant C (Group 2) | |

| 2D interface | 2D interface no template |
|---|---|
| (d) Participant D (Group 2) | |

Figure 14: Example gallery wall designs created by participants in our user evaluation pool.

The participant was then asked to create a gallery wall design that fits with the living room under a given condition. For **Condition 2D** and **Condition 2DNT**, the participant was presented with a photo of the living room when designing a gallery wall using the 2D interface (please refer to the supplementary material for a screenshot of the interface). Our tool tracked the interaction metrics for later analysis.

Figure 14 shows some gallery walls designed by the participants. Our supplementary material contains the results created by all participants.

## EXPERIMENT RESULTS

We discuss the user evaluation results with regard to performance, usage and user feedback. We use t-tests to evaluate if there is any significant difference between the results obtained under different pairs of conditions by reporting the p-values. We show our results in box plots for easy interpretation. Our supplementary material shows the numeric results and all designs created by participants under different conditions.

## Performance

We tracked the performance of retrieving items from the large database we used in our experiments on a desktop computer equipped with a i7–7700k 4.2GHz CPU, 16GB RAM, and an NVIDIA GeForce GTX 1080 8GB graphics card. Retrieving focal items took 3.12 seconds and retrieving auxiliary items took 1.21 seconds. All the other user interface operations ran at an interactive rate.

Our tool tracks the participant's performance under each given condition. Here are the performance metrics tracked:

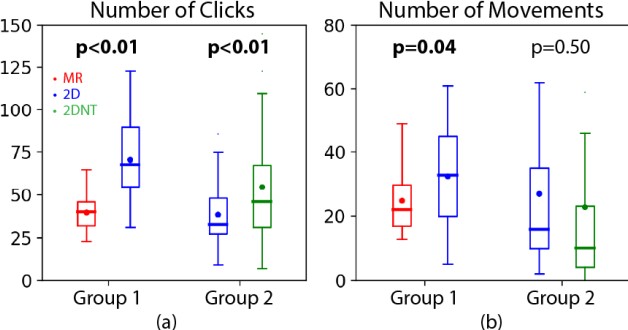

Figure 15: Performance results in different settings. Color dots and bars show the means and medians. The p-value of t-test computed between the results of the two conditions in each group is shown. The p-values smaller than 0.05 which reject the null hypothesis are bolded.

- **Number of Clicks:** The total number of times the participant clicked on a user interface component.

- **Number of Movements:** The total number of times the participant adjusted the position of an art item.

**Mixed Reality versus 2D Interface.** As the Group 1 results in Figure 15 show, under the MR condition where the participants designed via a Magic Leap One headset, they made fewer clicks (p<0.01 in t-test) and movements (p=0.04), compared to the 2D condition where they designed via a 2D interface.

Under the MR condition, the participants could see their design directly visualized on the real wall, which might result in fewer adjustments. In our perceptual study, we find that

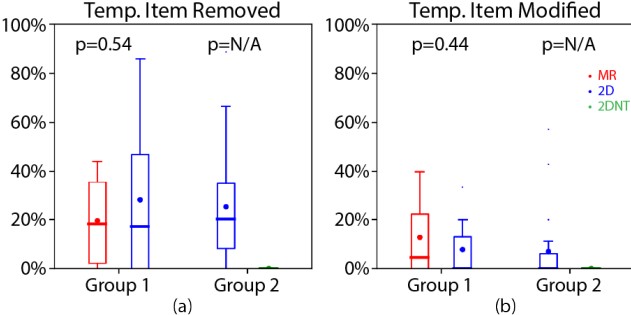

Figure 16: Usage statistics of the items placed by a template (Section 5.4) under different conditions.

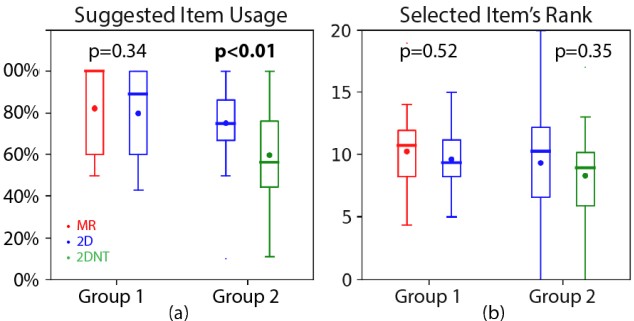

Figure 17: Usage statistics of the items suggested by the Item Panel (Section 6.3) under different conditions.

there is no significant difference between the visual quality of the designs created under the MR and the 2D conditions. In all, the direct visualization brought about by mixed reality allows the participants to create gallery wall designs of similar visual quality (compared to designs created on a 2D screen) with fewer manual adjustments.

**Templates versus No Templates.** As the Group 2 results in Figure 15(a) show, under the 2D condition where the participants created designs with the aid of templates they made fewer clicks (p<0.01) compared to the 2DNT condition where they created designs without the aid of templates. The template functionality helped them create designs more efficiently.

### Usage
Our tool also tracks the usage of the design suggestions generated by the Template Panel or Item Panel. Here are the metrics used:

- *Template Item Removed:* If a participant applied a template for initializing a gallery wall design, the percentage of items generated by the template that he/she removed.

- *Template Item Modified:* If a participant applied a template for initializing a gallery wall design, the percentage of items generated by the template that he/she modified.

- *Suggested Item Usage:* What percentage of items in the final gallery wall design was chosen among the top-20 suggestions from the Item Panel retrieved according to the user's specified criteria.

- *Selected Item's Rank:* The average rank of the items the participants selected from the suggestions of the Item Panel.

**Template Items.** Figure 16 shows the usage statistics of the items placed by a template under different conditions. As Figure 16(a) shows, the participants removed about 20% and 28% of the items placed by the templates under the MR and 2D conditions respectively. As a template places highly similar items in a design, it seems that the participants tended to replace a few items with other less similar items to introduce some variation or contrast into the overall design. On

the other hand, as Figure 16(b) shows, in average only about 13% and 8% of the items were modified under the MR and 2D conditions respectively. Overall, the participants kept a majority of the items placed by a template in their final gallery wall design.

**Suggested Items.** Figure 17 shows usage statistics of items suggested by the Item Panel. As Figure 17(a) shows, under the MR and 2D conditions, about 82% and 76% of the items used in the final design were chosen from the top-20 suggestions in the Item Panel retrieved according to their specified criteria. The Group 2 results show that there was a significant difference (p<0.01) between suggested item usages under the 2D and 2DNT conditions. Under the 2DNT condition when the participants could not use a template to initialize a gallery wall design, they tended to use items from the database more randomly.

On the other hand, as Figure 17(b) shows, the average rank of the selected items ranges from about 8 to 10 under different conditions. It seems that, in using the Item Panel, the participants tended to choose items that match with their specified criteria but not in a very strict sense.

### User Feedback
We talked to the participants after the experiments. They generally merited the visualization brought about by our mixed reality tool for showing the design on the real wall, which made the interactive design process more intuitive compared to a typical computer screen, as they could directly see how their design could fit with the real space. On the downside, some participants reported initial user experience challenges while acclimating themselves to the device's field-of-view and headset fit. We include the participants' comments in our supplementary material.

We also conducted a two-alternative forced-choice approach to evaluate the quality of the gallery wall designs created by the participants under two different conditions (*MR* & *2D* or *2D* & *2DNT*). We include the details of the perceptual study in the supplementary material.

### SUMMARY
We proposed a novel mixed reality-based interactive design tool for gallery walls. By overlaying a virtual gallery wall

on a real wall, our tool allows users to directly visualize their design in the real world as an integrated part of the creative process. Our suggestive design interface allows users to retrieve stylistically compatible items for creating their desired gallery walls.

### Limitations

We used a Magic Leap One headset for experimenting with our tool. The hardware still has limitations for consumer use. For example, the field-of-view is still a bit narrow for the user to see the overall design and the user interface at once; the user has to look around to see different things, which could be inconvenient.

It could be tiring to use the handheld controller for 3D interaction and manipulation in the 3D space for a prolonged period of time. Because of this, it would be challenging to use a sophisticated design tool that requires tedious and precise user input in mixed reality.

We believe that the visualization of the design in the real space by mixed reality may benefit the users in envisioning and communicating their designs, allowing them to design in the real space directly and intuitively. In the future, it would be worthwhile to extend our mixed reality design tool to consider more sophisticated context of the 3D scene to simplify other interior design tasks, such as suggesting furniture placement. Performing advanced scene analysis in real time for enabling interior design applications in mixed reality presents a practical challenge, which could be resolved as the computaional power of mixed reality devices continue to increase.

Due to the scope of this project, in this work we only focus on a subset of interior design, namely, gallery wall designs. We show that a mixed reality approach for gallery wall design is feasible. Our approach could be extended to consider 3D decorations (Figure 18), though they are less common and we did not consider them in our current approach.

It would be helpful to have a large database of gallery wall designs from which our tool could learn the spatial relationships and compatibility between different art items in a gallery wall design, and use such knowledge for synthesizing new designs.

### Future Work

With the advances of artificial intelligence and natural language processing techniques, we hope that such techniques can be adopted in mixed reality for enabling natural and convenient user interaction experiences in the interior design process. For example, it would be helpful if a mixed reality interior design software application could understand voice commands from the user to decorate a space, minimizing the need for manual user input. With such advances, using a mixed reality software application for interior design will be as natural as talking to an interior designer who can instantly visualize the design for the user through mixed reality. We believe this is an exciting goal for enabling human-AI collaboration in design.

For commercial purposes, art items could be further annotated with non-aesthetic tags, such as their prices, frequencies

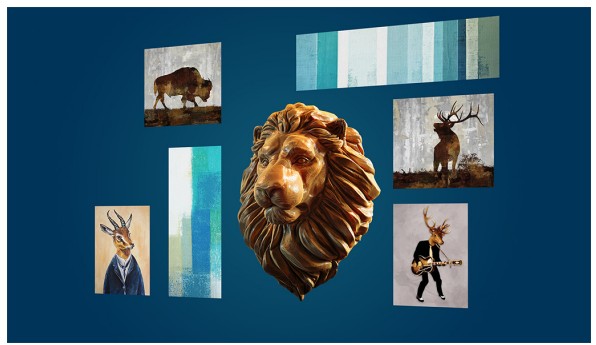

Figure 18: Example of a gallery wall design containing a 3D decoration object (a lion head) that can be visualized in mixed reality using our tool.

of being viewed or selected, and the semantic meanings they carry. Incorporating such additional tags could provide more desirable item recommendations while a user is creating their gallery wall designs.

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
