# OpenReview forum: "Interactive Design of Gallery Walls via Mixed Reality"
_graphicsinterface.org/Graphics_Interface/2020/Conference — Submitted to GI 2020_

### Official Review · AnonReviewer1 · 2020-04-19
**A report of a simple AR tool for interior gallery design, disproportion of content makes it hard to gauge where the technical or research contribution is.**

**Rating:** 4
**Confidence:** 4

**Review:**

This paper introduces an augmented-reality (AR) application in the form of a heads-up display helping both designers and novice users to create and visualize a gallery wall of art items (paintings), with suggestive features on spatial and color compatibility. It also reports a usability study evaluating the usage of the system.

I think this work proposes a niche feature (recommendation and application on interior design) to an already quite saturated arena of AR interior design tools (as mentioned in the Related Work, the idea of using AR to for viewing paintings is not new) and I'm not sure if it is enough for a full publication. There is about 1 page on the mathematics behind the art selection, and another 2 pages on the system functionalities. While informative, it further diminishes the research contribution of this paper. Especially when the technical approach doesn't seem to be a contribution this paper is making.

The evaluation is rather straight-forward and does not provide deeper insights on how the mixed reality mode of interior art gallery design helps the activity. One of the interview results from interior designers is it takes about 40 minutes to complete a conventional design process. Why time wasn't measured? And why not asking the designers in Group 1 more probing questions (the paper says "a company", I assume its the same design company)? I think their feedback (even with a few of them) would be useful for qualitative analysis and could be more insightful.

Overall, I think the content is disproportional to the contributions claimed by the authors. I'd expect reading a more in-depth qualitative analysis and reflection of the design than details of the system. My suggestion is to swap the supplementary materials with the technical approach, and condense the user interaction section.

Pros:
-Adequate coverage of related work.
-Writing is clear and easy to follow. Description of the implementation and user interaction is very detailed.
-It is really helpful to have most of the data included in the supplementary materials. Not many submissions do that.

Cons:
-Some clarifications needed:
--Workflow needs a bit more details. How does the user select one (the word "filter" was used but that's all I could find)? Base on what criteria is the color palette generated? Is it just one palette? Can the user freely adjust the weighs wc and ws?
--The 4 templates provided in the system seems to be limited to a "cluster" style. How about those common ones like an array of images, or art pieces of the same sizes side-by-side (the Align All function seems to do that but it's not clear how the user chooses between templates and other alignment functionalities)?

-A big portion of the paper is used to describe how arts are selected based on similarities and wall color. I understand it is useful to provide details like these. But they are not really the contributions claimed by the authors.
-Some of the sections feel repetitive. For example, the workflow and details of the functionalities, the description of the conditions. Consider combining each of them using subsections.

-Several shortcomings of the study methodology:
--The performance metrics (# clicks, # movements) need justification. I know these are common ones for task analysis, but why are they useful in interior design? One could argue more trial and errors could help generating better designs, which could result in more clicks and movements.
--The Usage evaluation reads more like the evaluation of the effectiveness of the template & suggestion, which seems to deviate from the focus of this paper.

Minor points:
-Some of the technical details can be moved to the workflow discussion for clarity. For example, the system retrieves the top-20 focal arts, that the user can choose between color and style compatibility when choosing auxiliary art items.
-I wonder if the user interface allows the user to only show the design canvas (without the other panels) so they can get a better sense of how the design will look like. The demo video doesn't really show how the mixed reality interface look like in action.
-Figure 11a is rather trivial for describing how "replace" works, perhaps it does more than just replacing an art item like auto-resizing?

---

### Official Review · AnonReviewer3 · 2020-04-20
**Useful tool, but a problematic presentation**

**Rating:** 4
**Confidence:** 4

**Review:**

The paper presents an AI powered mixed reality interface to compose image galleries.

The abstract and introduction could be improved.  Specifically, the problem could be introduced more carefully and a clear line of argumentation that highlights how the author’s solution, i.e., the mixed-reality gallery interface, addresses the outlined problem could be provided.

The related literature section suggests a lack in literature for gallery wall design. While this is not my area of expertise a database search reveals several paper that address virtual reality gallery design. Besides that just a lack in literature is not a sufficient motivation. Further, there has been plenty of work done on augmented and mixed reality for gallery design that there is no sufficient argument for a novel area. The background section focuses on interior design applications and automated layout design, which diverges from the introduction of the paper. See [1,2] for work on AR and galleries. Even if the work produced for art galleries is not what the authors intended to do, differentiating their work clearly from the work done before would be appreciated.

The approach, process, and design of the AI powered gallery recommender is the focal piece of the manuscript and sound in development. I am not an expert in machine learning, but the interface to use AR to modify and create wall galleries is well constructed. I can see that this part could be of interest to the GI community.

The user evaluation presents two groups, but within each group you find different conditions. I seems the others conducted two different studies. Considering that expertise or interest might affect the presented data, it would be good to know from what kind of company participants were recruited from. The generated insights are inconclusive. The confusion around the study design and the lack of insights make it very challenging to interpret the results from the user study. This section is not ready for publication.

In summary, I think the manuscript has too many flaws in the current stage. The introduction lacks clarity, the related work section is incomplete and partially disconnected from the introduction. The system is sound, but the user evaluation incomplete and could be better presented. I believe that the work can be turned into a valuable contribution, but , from my perspective, more work is required.


1.	Leue, M. C., Jung, T., & tom Dieck, D. (2015). Google glass augmented reality: Generic learning outcomes for art galleries. In Information and communication technologies in tourism 2015 (pp. 463-476). Springer, Cham.
2.	tom Dieck, M. Claudia, Timothy Hyungsoo Jung, and Dario tom Dieck. "Enhancing art gallery visitors’ learning experience using wearable augmented reality: generic learning outcomes perspective." Current Issues in Tourism 21, no. 17 (2018): 2014-2034.

---

### Official Review · AnonReviewer2 · 2020-04-21
**The paper presents an interactive design tool for creation of gallery walls in MR. The primary contribution is in the system design which is impressive. The evaluation is somewhat weak.**

**Rating:** 7
**Confidence:** 4

**Review:**

The paper has been written clearly. It addresses the specific problem of gallery wall design and how to ease the process for non-professional designers. This is an interesting problem and the paper presents an intuitive and effective solution.

It follows a user-centric design process, talking to designers first, that effectively motivates the design for the workflow. It further uses computational approaches to drive focal and auxiliary art suggestions. It then describes the interface features available to the users. The system design is methodical and it is delightful to read through.

The paper presents a unique contribution both from an MR perspective as well as from a design perspective in easing gallery wall design for regular users. However, I have a few questions and concerns -

1. The paper does not ground its decisions with references in multiple places. Some instances - a. Why select the neighbor colors and complementary colors? Are there any references that suggest that this is the optimal way to get compatible colors? b. Is the database public or described anywhere? c. Are there any references or literature on gallery wall clusters that refer to how they are always based on this idea of focal+auxiliary?

2. The user evaluation is surprisingly sparse in user's subjective feedback and usability scores. While it may be difficult to compare the MR usecase due to the FOV constraints and arm fatigue issues, some insights here will be very useful. As it stands right now, the evaluation does not inform much beyond showing that users are able to use all three interfaces, which is not saying much.

3. What were the instructions for time for the study? Were the users asked to finish each design within a particular time or were they asked to do the best job possible with flexible timings?

4. Why weren't tags used for auxiliary item suggestion, only color and visual features?

---

### Meta-Review · Area_Chair1 · 2020-04-23

**Recommendation:** Reject
**Confidence:** 4

**Metareview:**

All reviewers agreed that the design and implementation approach is sound and described in good details, but have different opinions on problem specifications and motivation: R2 believes those are presented effectively, while R3 believes those need more clarity and completeness. R1 also pointed out a disproportion between the presented content and claimed contributions.

Most reviewers expressed concerns about inclusion of sufficient related work, so as to ground design decision (R2) and differentiate with prior work (R3).

A main weakness pointed out by all reviewers is that the evaluation is lacking, that it is sparse (R2) and does not provide enough insights (R1&3).

At its current stage I think the paper is not ready for publication, but could be improved by stronger linkage with related work (R2&3), providing clarification in study procedures (R1,2&3), and a more in-depth evaluation to provide more information and insight (R1,2&3).

---

### Decision · Program_Chairs · 2020-04-25

Reject